# Infra-Red Active Dirac Plasmon Serie in Potassium Doped-Graphene (KC_8_) Nanoribbons Array on Al_2_O_3_ Substrate

**DOI:** 10.3390/ma14154256

**Published:** 2021-07-30

**Authors:** Josip Jakovac, Leonardo Marušić, Denise Andrade-Guevara, Julio C. Chacón-Torres, Vito Despoja

**Affiliations:** 1Institut za Fiziku, Bijenička 46, 10000 Zagreb, Croatia; jjakovac@ifs.hr; 2Maritime Department, University of Zadar, M. Pavlinovića 1, 23000 Zadar, Croatia; lmarusic@unizd.hr; 3School of Physical Sciences and Nanotechnology, Yachay Tech University, Urcuquí 100119, Ecuador; denise.guevara@gmail.com (D.A.-G.); julio.chacon@gmail.com (J.C.C.-T.); 4Donostia International Physics Center (DIPC), P. Manuel de Lardizabal, 4, 20018 San Sebastián, Spain

**Keywords:** graphene ribbons, 2D plasmons, photonics, intercalated graphene

## Abstract

A theoretical formulation of the electromagnetic response in graphene ribbons on dielectric substrate is derived in the framework of the ab initio method. The formulation is applied to calculate the electromagnetic energy absorption in an array of potassium-doped graphene nanoribbons (KC${}_{8}$-NR) deposited on a dielectric Al${}_{2}$O${}_{3}$ substrate. It is demonstrated that the replacement of the flat KC${}_{8}$ by an array of KC${}_{8}$-NR transforms the Drude tail in the absorption spectra into a series of infrared-active Dirac plasmon resonances. It is also shown that the series of Dirac plasmon resonances, when unfolded across the extended Brillouin zones, resembles the Dirac plasmon. The Dirac plasmon resonances’ band structure, within the first Brillouin zone, is calculated. Finally, an excellent agreement between the theoretical absorption and recent experimental results for differential transmission through graphene on an SiO${}_{2}$/Si surface is presented. The theoretically predicted micrometer graphene nanoribbons intercalation compound (GNRIC) in a stage-I-like KC${}_{8}$ is confirmed to be synthesized for Dirac plasmon resonances.

## 1. Introduction

Recently, graphene-based plasmonics and photonics are being used in various applications, such as bio-chemical sensing enhancement [1,2], photovoltaic efficiency enhancements or amplification of graphene photoemission [3,4], in optoelectronics in the THz and the infrared (IC) frequency region and in spintronics [5,6]. Graphene plasmonics are also being tested for use in telecommunications [7,8,9,10,11,12]. Graphene-based gas detectors use localized plasmons in the graphene nanoribbons to identify the rotational-vibrational modes in various gas molecules [13].

One of the biggest challenges in applied plasmonics or photonics is finding a way to excite and manipulate the 2D plasmons directly by the incident electromagnetic field. Even though 2D plasmons modes produce a strong localized electric field, that field is evanescent and therefore cannot be excited directly by light. In single-layer graphene, the Dirac plasmon can be excited only indirectly, e.g., by exciting the localized plasmons on the AFM tip, which then excites the Dirac plasmon in the graphene [14]. However, subwavelength nanostructures such as graphene nanoribbons (GNRs) support ‘plasmon resonances’ with very localized electric field that can be radiated into the surrounding area; thus, it can also be pumped directly by an external radiation.

The measurements of the electromagnetic field transmission through the GNR arrays on SiO${}_{2}$/Si substrate clearly show the existence of strong plasmon resonances in the THz and mid-infrared frequency range (depending on nanoribbon thickness) [15,16]. Moreover, the infrared near-field imaging of GNRs on an Al${}_{2}$O${}_{3}$ substrate shows that, in addition to the conventional plasmonic resonances, GNRs support the edge plasmons (distributed along the graphene edges [17]). The plasmon resonances in submicrometer multilayer graphene ribbons on an Si/SiO${}_{2}$ substrate caused by different doping concentrations (graphene Fermi energy) enable the tuning of the IR reflectivity [18]. In the above experiments, the plasmon resonances appear in the THz and IR frequency ranges, thus hybridization with IR active Fuchs–Kliewer surface optical (SO) phonons [19] in polar substrates was also considered.

Thus far, the theoretical description of the optically active plasmon resonances in submicrometer graphene ribbons, despite providing useful information, such as plasmon wave functions or analytical dispersion relations, is limited to semi-analytical modeling, mostly based on the simple Drude model conductivity, which does not take into account the substrate polarization [20,21]. The ab initio calculations of the energy loss function in semimetallic (zigzag) and semiconducting (armchair) nanoribbons [22] provide information about the interesting interplay between the intraband and interband plasmons. The ab initio calculations of the dielectric response in different GNRs, taking into account the electron scattering with SO phonons in various polar and nonpolar substrates, provided very useful information about plasmon propagation length and plasmon–phonon hybridization [23]. However, these ab initio studies have only focused on a few nanometers thick GNRs, while the radiative plasmon resonances were not studied. Interesting experimental and theoretical studies show that the THz absorbance of the graphene monolayer can be considerably enhanced by depositing the graphene on a dielectric substrate of specific dielectric permittivity and thickness [24,25]. Another very interesting phenomenon of the modulation of the THz graphene absorption is achieved by applying an optical pump signal, which modifies the conductivity of the graphene sheet [25]. However, the sharp plasmon resonances, which occur only in graphene ribbons, are not observed in these investigations.

A way of approaching the creation of a Dirac plasmon (DP) in GNRs has been explored via the synthesis of alkali-metal-doped graphene on metallic substrates, being extensively studied experimentally and theoretically [26,27,28,29]. These experiments have shown that, by doping graphene with electron donors, the Dirac plasmon resonances can be excited and extensively studied. Furthermore, the use of advanced multilayer graphene nanoribbons will help control the plasmonic resonances derived from the perpendicular electric field in those nanostructures [18].

In this paper, we explore the electromagnetic response in an array of potassium-doped graphene nanoribbons (KC${}_{8}$-NR) deposited on a (Al${}_{2}$O${}_{3}$) dielectric substrate. The single-layer KC${}_{8}$ (KC${}_{8}$-SL) optical conductivity tensor $\sigma_{\mu \nu }^{0}\left(\omega \right)$ and the bulk Al${}_{2}$O${}_{3}$ macroscopic dielectric function $\epsilon_{s}\left(\omega \right)$ are calculated from first principles. Special attention is paid to the series of Dirac plasmon resonances (DPR) in d=50, 100 and 200 nm thick KC${}_{8}$-NR arrays. We show that the series of DPR consists of a series of dipolar or infrared-active DPR and a series of non-dipolar or dark DPR. We demonstrate that the DPR in the first Brillouin zone (1stBZ) when unfolded in the extended Brillouin zone (exBZ) resembles the Dirac plasmon (DP) in the KC${}_{8}$-SL. For smaller separations between the nanoribbons, thus with a stronger interaction between, which causes dispersion, we calculate the resulting DPR band structure within the 1stBZ. Finally, we apply the proposed formulation to calculate the electromagnetic absorption in the doped graphene microribbons on the SiO${}_{2}$/Si surface. The results are then compared with experimental measurements of differential transmission through the same sample [15], and KC${}_{8}$ micrometer length GNRs intercalation compounds were synthesized.

The rest of the paper is organized as follows. In Section 2, we present the theoretical model used to calculate the electromagnetic energy absorption A in the array of KC${}_{8}$-NR. In Section 3, we present the ab initio computational details and the results for the KC${}_{8}$-SL optical conductivity $\sigma_{yy}^{0}\left(\omega \right)$ and the bulk Al${}_{2}$O${}_{3}$ macroscopic dielectric function $\epsilon_{s}\left(\omega \right)$. In Section 4, we present the results for the absorption spectra A in different arrays of KC${}_{8}$-NR and DPR band structure. In Section 5, we show the comparison with available experimental results. Section 6 contains some conclusive remarks. Unless stated otherwise, atomic units are used, i.e., e=ℏ=m=1, where *e* is the electronic charge, *ℏ* is the reduced Plank constant, *m* is the electron mass and *c* is the speed of light in vacuum.

## 2. Theoretical Formulation

### 2.1. Calculation of the Electromagnetic Energy Absorption A

In this section, we briefly describe the method of calculation of the electromagnetic energy absorption in the system consisting of an array of KC${}_{8}$-NR deposited on a dielectric Al${}_{2}$O${}_{3}$ substrate (according to the formulation developed by [30]). The dielectric substrate occupies the region z<0, while the KC${}_{8}$-NR of width *d* and period *l* are arranged so that their graphene layers are placed at $z=z_{0}$ above the dielectric surface, as illustrated in Figure 1. The substrate polarization is described by the dielectric function $\epsilon_{s}$, while the polarization of the dielectric media occupying the region z>0 is described by the dielectric function $\epsilon_{0}$.

If we assume that the sample is driven by an external electromagnetic field of unit amplitude, frequency ω and wave vector *k*, with incidence perpendicular to the surface,
(1)E=ecos(kz−ωt).
the polarization e is parallel to the surface and ω=kc, where *c* is the speed of light. The electromagnetic energy absorption in the array of KC${}_{8}$-NR can be obtained by using the following expression [30]:(2)$$A\left(\omega \right)=\sum_{\mu \nu =x,y,z}e_{\mu }e_{\nu }ℜ\int drdr^{\prime }e^{-ikz}\;\sigma_{\mu \nu }\left(r,r^{\prime },\omega \right)\;e^{ikz^{\prime }}.$$

Considering that the thickness of the KC${}_{8}$-SL is significantly smaller than the IR or visible light wavelength ($\lambda =\frac{2\pi c}{\omega }>100$ nm), and taking into account only the bare polarizations (μ=x or *y*), Equation (Equation 2) can be simplified as
(3)$$A\left(\omega \right)=ℜ\int drdr^{\prime }\sigma_{\mu \mu }\left(r,r^{\prime },\omega \right).$$

Here, the tensor $\sigma_{\mu \nu }$ represents the screened conductivity of the KC${}_{8}$-NR array, which is the solution of the Dyson equation [31]:(4)$$\begin{array}{c} \sigma_{\mu \nu }\left(r,r^{\prime },\omega \right)=\sigma_{\mu \nu }^{0}\left(r,r^{\prime },\omega \right)+\sum_{\alpha \beta =x,y,z}\int dr_{1}dr_{2}\;\sigma_{\mu \alpha }^{0}\left(r,r_{1},\omega \right)\Gamma_{\alpha \beta }\left(r_{1},r_{2},\omega \right)\sigma_{\beta \nu }\left(r_{2},r^{\prime },\omega \right), \end{array}$$
where $\sigma_{\mu \nu }^{0}$ is the nonlocal irreducible conductivity tensor of the KC${}_{8}$-NR array and $\Gamma_{\alpha \beta }$ is the propagator of the bare electric field corrected by the presence of the Al${}_{2}$O${}_{3}$ substrate. The part of the electromagnetic energy absorbed by the Al${}_{2}$O${}_{3}$ substrate is neglected, which is a reasonable approximation considering that the Al${}_{2}$O${}_{3}$ is mostly transparent in the frequency interval of interest. Both tensors, $\sigma^{0}$ and Γ, are described explicitly below.

If the nanoribbon width is much larger than the unit cell constant (d>>a), the nonlocal effects in the x−y plane are negligible and the optical response of the KC${}_{8}$-NR can be approximated by the local optical conductivity. Moreover, since the thickness of the KC${}_{8}$-SL is significantly smaller than the wavelength λ>100 nm, it can be treated as a 2D crystal, localized, e.g., in the graphene, $z=z_{0}$ plane. In this approximation, the conductivity tensor becomes:(5)$$\sigma_{\mu \nu }^{0}\left(r,r^{\prime },\omega \right)\approx \delta \left(z-z_{0}\right)\sigma_{\mu \nu }^{0}\left(\rho ,\omega \right)\delta \left(r-r^{\prime }\right),$$
where ρ=(x,y) is the 2D position vector,
(6)$$\begin{array}{c} \sigma_{\mu \nu }^{0}\left(\rho ,\omega \right)=\sigma_{\mu \nu }^{0}\left(\omega \right)\sum_{n=-\infty }^{n=\infty }\left[\theta (y-nl+d/2)-\theta (y-nl-d/2)\right], \end{array}$$
and $\sigma_{\mu \nu }^{0}\left(\omega \right)$ is the 2D optical conductivity of the KC${}_{8}$-SL. All this enables the Fourier expansion of $\sigma_{\mu \nu }^{0}$:(7)$$\begin{array}{c} \sigma_{\mu \nu }^{0}\left(r,r^{\prime },\omega \right)=\delta \left(z-z_{0}\right)\delta \left(z-z^{\prime }\right)\sum_{gg^{\prime }}\int \frac{dQ}{{\left(2\pi \right)}^{2}}\sigma_{\mu \nu ,gg^{\prime }}^{0}\left(\omega \right)e^{i(Q+G)\rho }e^{-i\left(Q+G^{\prime }\right)\rho^{\prime }}. \end{array}$$

Here, the 2D reciprocal vectors are G=(0,g), with $g=\frac{2\pi n}{l};\;\;n=0\pm 1,\pm 2,\ldots$, and
(8)$$\sigma_{\mu \nu ,gg^{\prime }}^{0}\left(\omega \right)=\sigma_{\mu \nu }^{0}\left(\omega \right)\left\{\begin{array}{cc} \frac{2}{l(g-g^{\prime })}\sin\left[\left(g-g^{\prime }\right)d/2\right]; & \;g\neq g^{\prime }\\ d/l;\;\;g=g^{\prime } \end{array}\right..$$
where $Q=(Q_{x},Q_{y})$ is the 2D transfer wave vector.

The propagator Γ remains translationally invariant in the x−y direction so it can be Fourier transformed as
(9)$$\begin{array}{c} \Gamma_{\mu \nu }\left(r,r^{\prime },\omega \right)=\sum_{gg^{\prime }}\int \frac{dQ}{{\left(2\pi \right)}^{2}}\Gamma_{\mu \nu ,gg^{\prime }}\left(Q,\omega ,z,z^{\prime }\right)e^{i(Q+G)\rho }e^{-i\left(Q+G^{\prime }\right)\rho^{\prime }}, \end{array}$$
where
(10)$$\Gamma_{\mu \nu ,gg^{\prime }}\left(Q,\omega ,z,z^{\prime }\right)=\Gamma_{\mu \nu }\left(Q+G,\omega ,z,z^{\prime }\right)\delta_{gg^{\prime }}.$$

Using the expansions (Equation 7) and (Equation 9) and assuming that the screened conductivity σ can be transformed the same way as the conductivity $\sigma^{0}$ (expansion Equation (Equation 7)), the Dysons Equation (Equation 4) transforms into matrix equation for the screened conductivity
(11)$$\begin{array}{c} \sigma_{\mu \nu ,gg^{\prime }}\left(Q,\omega \right)\;=\sigma_{\mu \nu ,gg^{\prime }}^{0}\left(Q,\omega \right)+\sum_{\alpha \beta }\sum_{g_{1}g_{2}}\sigma_{\mu \alpha ,gg_{1}}^{0}\left(Q,\omega \right)\Gamma_{\alpha \beta ,g_{1}g_{2}}\left(Q,\omega ,z_{0},z_{0}\right)\sigma_{\beta \nu ,g_{2}g^{\prime }}\left(Q,\omega \right). \end{array}$$

After inserting the Fourier expansion of the screened conductivity (Equation (Equation 7), where $\sigma^{0}\to \sigma$) into Equation (Equation 3), and using the identity $\sigma_{\mu \mu ,g-g_{0}g^{\prime }-g_{0}}\left(Q+G_{0},\omega \right)=\sigma_{\mu \mu ,gg^{\prime }}\left(Q,\omega \right)$, we obtain the final expression for the electromagnetic energy absorption rate rate per unit area
(12)$$A\left(\omega \right)=ℜ\sigma_{\mu \mu ,g=0g^{\prime }=0}\left(Q=0,\omega \right).$$

We use the expression (Equation 12) to determine the intensity of the electromagnetic modes beyond the optical limit as well, e.g., in the nonradiative or evanescent region (Q>ω/c), simply by using the conductivity σ calculated for a finite wavevector (Q≠0).

### 2.2. Electric Field Propagator

The propagator of the electric field can be written as
(13)$$\hat{\Gamma }={\hat{\Gamma }}^{0}+{\hat{\Gamma }}^{sc},$$
where the propagator of the ‘free’ electric field (or free photon propagator) is [31,32]
(14)$$\begin{array}{c} {\hat{\Gamma }}^{0}\left(Q,\omega ,z,z^{\prime }\right)=-\frac{4\pi i}{\epsilon_{0}\omega }\delta \left(z-z^{\prime }\right)z\cdot z-\;\\ \frac{2\pi \omega }{\beta_{0}c^{2}}e^{i\beta_{0}\left|z-z^{\prime }\right|}\sum_{q=s,p}e_{q}^{0}\cdot e_{q}^{0}. \end{array}$$

The propagator of the scattered electric field in the region z>0 is [32]
(15)$${\hat{\Gamma }}^{sc}\left(Q,\omega ,z,z^{\prime }\right)=-\frac{2\pi \omega }{\beta_{0}c^{2}}e^{i\beta_{0}\left(z+z^{\prime }\right)}\sum_{q=s,p}r_{q}\cdot e_{q}^{+}\cdot e_{q}^{-}.$$

Here, the unit vectors of the s(TE) polarized electromagnetic field are $e_{s}^{0,\pm }=Q_{0}\times z$. The unit vectors of p(TM) polarized electromagnetic field are $e_{p}^{0,\pm }=\frac{c}{\omega \sqrt{\epsilon_{0}}}\left[\alpha_{0,\pm }\beta_{0}Q_{0}+Qz\right]$, where $\alpha_{0}=-sgn\left(z-z^{\prime }\right)$, $\alpha_{\pm }=\mp 1$ and $Q_{0}$ and z are the unit vectors in the Q and *z* directions, respectively. More specifically, $\Gamma^{sc}$ represents the electric field produced by an external point dipole which is reflected at the dielectric surface. Therefore, ‘−’ represents the incident electric field, while ‘+’ represents the reflected electric field. $\Gamma^{0}$ represents the ‘direct’ electrical field produced by point the dipole so that the superscript ‘0’ represents the spherical forward propagating field. The reflection coefficients of the s(TE) and p(TM) polarized electromagnetic waves at the media/substrate interface are $r_{s}=\left(\beta_{0}-\beta_{s}\right)/\left(\beta_{0}+\beta_{s}\right)$ and $r_{p}=\left(\beta_{0}\epsilon_{s}-\beta_{s}\epsilon_{0}\right)/\left(\beta_{0}\epsilon_{s}+\beta_{s}\epsilon_{0}\right)$, respectively. The complex wave vectors in the perpendicular (*z*) direction are $\beta_{0,s}=\sqrt{\frac{\omega^{2}}{c^{2}}\epsilon_{0,s}\left(\omega \right)-Q^{2}}$.

### 2.3. Calculation of RPA Optical Conductivity $\sigma_{\mu \nu }\left(\omega \right)$

Since we study a 2D crystal which consists of just two atomic layers, its electromagnetic response is strongly dispersive in the perpendicular *z* direction. For this reason, we define the spatially dependent conductivity
$$\sigma_{\mu \nu }^{0}\left(Q,\omega ,z,z^{\prime }\right)=\frac{1}{L}\sum_{G_{z}G_{z}^{\prime }}\sigma_{\mu \nu ,G_{z}G_{z}^{\prime }}^{0}\left(Q,\omega \right)e^{iG_{z}z-iG_{z}^{\prime }z^{\prime }},$$
where the conductivity matrix is defined as [30]
(16)$$\begin{array}{c} \sigma_{\mu \nu ,G_{z}G_{z}^{\prime }}^{0}\left(Q,\omega \right)=-\frac{i\hbar }{\Omega }\sum_{K,n,m}\frac{1}{E_{nK}-E_{mK+Q}}\frac{f_{nK}-f_{mK+Q}}{\hbar \omega +i\eta +E_{nK}-E_{mK+Q}}\times \;\\ j_{nK,mK+Q}^{\mu }\left(G_{z}\right)\;{\left[j_{nK,mK+Q}^{\nu }\left(G_{z}^{\prime }\right)\right]}^{*}. \end{array}$$

Here, the current vertices are
(17)$$j_{nK,mK+Q}^{\mu }\left(G_{z}\right)\;=\;\int_{\Omega }dre^{-iQ\rho }e^{-iG_{z}z}\;j_{nK,mK+Q}^{\mu }\left(r\right),\;$$
and the current produced by the transitions between the Bloch states $\phi_{nK}^{*}\to \phi_{mK+Q}$ is defined as
$$\begin{array}{c} j_{nK,mK+Q}^{\mu }\left(r\right)\;=\frac{e\hbar }{2im}\left\{\phi_{nK}^{*}\left(r\right)\partial_{\mu }\phi_{mK+Q}\left(r\right)-\left[\partial_{\mu }\phi_{nK}^{*}\left(r\right)\right]\phi_{mK+Q}\left(r\right)\right\}, \end{array}$$
where $G_{z}=2\pi n/L;n\in Z$ represents the reciprocal vector in the *z* direction, $K=(K_{x},K_{y})$ is the 2D wave vector and $\phi_{nK}$ and $E_{nK}$ are the Bloch wave functions and energies obtained by the DFT calculations. The spin quantum number ‘*s*’ is merged with the band quantum number, i.e., n≡(n,s), Ω=S×L is the normalization volume, *S* is the normalization surface, *L* is the superlattice unit cell in the *z* direction and $f_{nK}={\left[e^{(E_{nK}-E_{F})/kT}+1\right]}^{-1}$ is the Fermi–Dirac distribution at temperature *T*. $\eta_{intra}$ and $\eta_{inter}$ represent the phenomenological intraband and interband damping parameters, respectively. The two-dimensional conductivity used in this study is defined as
$$\begin{array}{c} \sigma_{\mu \nu }^{0}\left(\omega \right)=\int_{-L/2}^{L/2}dzdz^{\prime }\sigma_{\mu \nu }^{0}\left(Q=0,\omega ,z,z^{\prime }\right)\;=\;L\sigma_{G_{z}=0{G_{z}}^{\prime }=0}^{0}\left(Q=0,\omega \right). \end{array}$$

It should be noted that this defined conductivity do not depend on lattice parameter *L*. The KC${}_{8}$-SL is a conductive 2D crystal so it is appropriate to divide its RPA optical conductivity into intraband and interband contributions
(18)$$\sigma_{\mu \nu }^{0}\left(\omega \right)=\sigma_{\mu \nu }^{\mathrm{intra}}\left(\omega \right)\;+\;\sigma_{\mu \nu }^{\mathrm{inter}}\left(\omega \right),$$
which are both determined from the optical limit of the nonlocal conductivities $\sigma_{\mu \nu }^{i}\left(\omega \right)=\sigma_{\mu \nu }^{i}\left(\omega ,Q\approx 0\right)$. According to (Equation 16), the nonlocal intraband (n=m) conductivity is [30]
(19)$$\sigma_{\mu \nu }^{\mathrm{intra}}\left(\omega \right)=i\frac{e^{2}}{m}\;\frac{n_{\mu \nu }}{\omega +i\eta_{intra}},$$
where the effective number of the charge carriers is
(20)$$n_{\mu \nu }\;=\;-\frac{m}{Se^{2}}\;\sum_{n}\sum_{K\in 1.\mathrm{SBZ}}\;\frac{\partial f_{nK}}{\partial E_{nK}}\;j_{nK,nK}^{\mu }{\left[j_{nK,nK}^{\nu }\left(G_{z}=0\right)\right]}^{*}.$$

Here, K∈1.SBZ indicates that summation is performed within the first surface Brillouin zone. The nonlocal interband (n≠m) conductivity is [30]
(21)$$\begin{array}{c} \sigma_{\mu \nu }^{\mathrm{inter}}\left(Q,\omega \right)\;=\;-\frac{i\hbar }{S}\sum_{n\neq m}\sum_{K\in 1.SBZ.}\frac{j_{nK,mK+Q}^{\mu }\left(G_{z}=0\right){\left[j_{nK,mK+Q}^{\nu }\left(G_{z}^{\prime }=0\right)\right]}^{*}}{E_{nK}-E_{mK+Q}}\times \;\\ \frac{f_{nK}-f_{mK+Q}}{\hbar \omega +i\eta_{intra}+E_{nK}-E_{mK+Q}}. \end{array}$$

An alternative modeling of the KC${}_{8}$-SL conductivity could be done in analogy with the graphene conductivity modeling in the Dirac cone approximation [33], but taking into account the parabolic K(σ) band crossing the Fermi level.

### 2.4. Calculation of Substrate Macroscopic Dielectric Function $\epsilon_{s}\left(\omega \right)$

We assume that the dielectric media is vacuum (i.e., $\epsilon_{0}=1$) and that the substrate is aluminium-oxide (Al${}_{2}$O${}_{3}$) described by the macroscopic dielectric function $\epsilon_{s}\left(\omega \right)$. To calculate $\epsilon_{s}\left(\omega \right)$, we start from the 3D Fourier transform of the independent electron response function
(22)$$\begin{array}{c} \chi_{GG^{\prime }}^{0}\left(q,\omega \right)=\frac{2}{\Omega }\sum_{k\in 1.\mathrm{BZ}}\sum_{n,m}\;\frac{f_{n}\left(k\right)-f_{m}\left(k+q\right)}{\omega +i\eta +E_{n}\left(k\right)-E_{m}\left(k+q\right)}\rho_{nk,mk+q}\left(G\right)\;\rho_{nk,mk+q}^{*}\left(G^{\prime }\right), \end{array}$$
where k∈1.BZ indicates that summation is provided within first Brillouin zone. The charge vertices are defined as
(23)$$\rho_{nk,mk+q}\left(G\right)=\int_{\Omega }dr\;\phi_{nk}^{*}\left(r\right)e^{-i(q+G)r}\phi_{mk+q}\left(r\right).$$

Here, $k=(k_{x},k_{y},k_{z})$, $q=(q_{x},q_{y},q_{z})$ and $G=(G_{x},G_{y},G_{z})$ are the 3D wave vector, the transfer wave vector and the reciprocal lattice vector, respectively, and the integration is performed over the normalization volume Ω. We use the response matrix (Equation 22) to determine the dielectric matrix as
(24)$$E_{GG^{\prime }}\left(q,\omega \right)=\delta_{GG^{\prime }}-\sum_{G_{1}}v_{GG_{1}}\left(q\right)\chi_{G_{1}G^{\prime }}^{0}\left(q,\omega \right),$$
where the bare Coulomb interaction is $v_{GG^{\prime }}\left(q\right)=\frac{4\pi }{{\left|q+G\right|}^{2}}\;\delta_{GG^{\prime }}$. Finally, the macroscopic dielectric function is determined by inverting the dielectric matrix
(25)$$\epsilon_{s}\left(\omega \right)=\epsilon_{1}\left(\omega \right)+i\epsilon_{2}\left(\omega \right)=1/E_{G=0G^{\prime }=0}^{-1}\left(q\approx 0,\omega \right).$$

## 3. Computational Details

The KS wave functions $\phi_{nK}$ and energies $E_{nK}$ used to calculate the RPA conductivities $\sigma_{\mu \nu }$ and the substrate macroscopic dielectric function ϵ(ω) are determined using the plane-wave self-consistent field DFT code (PWSCF) within the QUANTUM ESPRESSO (QE) package [34]. For all crystal structures (KC${}_{8}$-SL, doped graphene and bulk Al${}_{2}$O${}_{3}$), the core-electrons interaction is approximated by the norm-conserving pseudopotentials [35,36]. The exchange correlation (XC) potentials in the KC${}_{8}$-SL and Al${}_{2}$O${}_{3}$ are approximated by the Perdew–Burke–Ernzerhof (PBE) generalized gradient approximation (GGA) functional [37] and in the graphene by the Perdew–Zunger local density approximation (LDA) functional [38]. The ground state electronic density in KC${}_{8}$-SL is calculated using the 8×8×1 Monkhorst–Pack K-mesh [39], the plane-wave cut-off energy is 60Ry and we use the hexagonal Bravais lattice, where a=4.922 Å and the separation between the KC${}_{8}$ layers is L=2.5a. Since the graphene unit cell is doped by holes, with the doping concentration $1.5\times 10^{13}$ cm${}^{-2}$, the graphene ground state electronic density is calculated using a dense 101×101×1 K-mesh and the plane-wave cut-off energy 60Ry. The Bravais lattices is hexagonal, where a=2.461 Å and the separation between the graphene layers is L=5a. The ground state electronic density of the bulk Al${}_{2}$O${}_{3}$ is calculated using 9×9×3 K-mesh, the plane-wave cut-off energy is 50Ry and the Bravais lattices is hexagonal (12 Al and 18 O atoms in the unit cell) with the lattice constants a=4.76 Å and c=12.99 Å.

The optical conductivity (Equation 18)–(Equation 21) in the KC${}_{8}$-SL is calculated using a 201×201×1 K-mash and the band summations (n,m) are performed over 100 bands. The damping parameters are $\eta_{intra}=10$ meV and $\eta_{inter}=40$ meV and the temperature is T=25 meV. The graphene optical conductivity is calculated using a 601×601×1 K-mash and the band summations are performed over 20 bands. The damping parameters are $\eta_{intra}=1$ and 15 meV, $\eta_{intra}=25$ meV and the temperature is T=25 meV. The response function (Equation 22) of the Al${}_{2}$O${}_{3}$ is calculated using a 21×21×7*k*-point mesh and the band summations (n,m) are performed over 120 bands. The damping parameter is η=100 meV and the temperature is T=10 meV. For the optically small wave vectors q≈0 used in this modeling, the crystal local field effects are negligible, so the crystal local field effects cut-off energy is set to zero.

Figure 2 shows the ab initio optical conductivity $ℜ\sigma_{yy}^{0}$ in the KC${}_{8}$-SL. The intraband contribution $\sigma^{\mathrm{intra}}$ is turquoise shaded, while the interband contribution $\sigma^{\mathrm{inter}}$ is orange shaded. The two pronounced peaks at ω≈4 and 14 eV correspond with the interband transitions between the graphene C(π) and C(σ) bands. The insert in Figure 2 shows the KC${}_{8}$ band structure, and we can see that the KC${}_{8}$ band structure does not differ much from the graphene band structure. The only influence of the K adatoms is the appearance of the potassium parabolic K(σ) band (turquoise dashed lines show the parabolic fit of the K(σ) band for the effective mass $m^{*}=0.92$) which abundantly donates electrons to the graphene C(π) band (denoted by magenta dashed lines) but in the way that it still remains partially filled. This causes the Fermi level shift by 1 eV above the Dirac point so that the onset for the interband transitions between the graphene C(π) bands appear at 2 eV (also denoted by brown dashed line in Figure 2). At the same time, this causes the appearance of two intraband excitations channels, K(σ) and C(π), which appear as strong Drude peak (shaded by turquoise color at ω≈0). Accordingly, this provides large effective number of charge carriers, $n_{yy}=0.021a_{0}^{-2}[7.58\times 10^{14}$ cm${}^{-2}$], resulting in a very intensive Dirac plasmon with zero direct interband damping.

Figure 3a shows the ab initio macroscopic dielectric function (Equation 25) of the bulk Al${}_{2}$O${}_{3}$ crystal. We can see that $\epsilon_{1}$ is almost constant ($\epsilon_{1}\approx 3$) for low frequencies (ω<3 eV), i.e., in the IR and even in the visible range, while $\epsilon_{2}$ is zero up to the band gap energy ($E_{g}$∼6 eV). This suggests that Al${}_{2}$O${}_{3}$ is a good choice for the substrate for the IR plasmonics, since its electronic excitations are far above the IR plasmons, and its IR active SO phonons (at $\omega_{TO}<100$ meV) [40] are still below the IR plasmons considered here. Therefore, in the frequency range of interest (red frame in Figure 3a), there is no dissipation of the electromagnetic energy in the substrate (it is transparent) and the dielectric function is constant.

Finally, in Figure 3b, we demonstrate the influence of the KC${}_{8}$-SL interband transitions and the influence of the dielectric substrate polarization on the Dirac plasmon dispersion relations. The dispersion relations are derived from the maxima of the real part of the screened conductivity
(26)$$\sigma_{yy}\left(Q,\omega \right)=\frac{\sigma_{yy}^{0}}{1-\Gamma_{yy}\left(Q,\omega \right)\sigma_{yy}^{0}\left(\omega \right)}$$
for $\sigma^{0}=\sigma^{\mathrm{intra}}$ and $\Gamma =\Gamma^{0}$ (brown dashed line), $\sigma^{0}=\sigma^{\mathrm{intra}}+\sigma^{\mathrm{inter}}$ and $\Gamma =\Gamma^{0}$ (orange dashed-dotted line) as well as $\sigma^{0}=\sigma^{\mathrm{intra}}+\sigma^{\mathrm{inter}}$ and $\Gamma =\Gamma^{0}+\Gamma^{sc}$ (solid turquoise line). We can see that the interband transitions significantly push the Dirac plasmon towards the lower frequencies. The substrate additionally screens the Dirac plasmon (reduces its energy), which is especially important in the optical region (Q<0.1 nm${}^{-1}$) when it deviates from the standard square-root behavior in the self-standing sample.

## 4. Results

In this section, we first demonstrate how replacing the KC${}_{8}$-SL by the KC${}_{8}$-NR of various widths *d* influences the absorption spectra A. Then, we demonstrate that the series of the Dirac plasmon resonances in the KC${}_{8}$-NR when unfold in ExBZ resembles the Dirac plasmon in the KC${}_{8}$-NR. Finally, we present the DPR band structure within the 1stBZ. We focus on the $e=\hat{y}$ (perpendicular to the NR) polarized electromagnetic field, and the separation between the graphene planes and the dielectric surfaces is fixed to $z_{0}=3.0$ Å.

The blue lines in Figure 4 show the absorption spectra (Equation 12) of the KC${}_{8}$-NR arrays of widths: (a) d = 50 nm; (b) d = 100 nm; (c) d = 200 nm. The dashed orange lines show the absorption spectra of the KC${}_{8}$-SL. Both structures are deposited on a dielectric model Al${}_{2}$O${}_{3}$ surface and the nanoribbon period is chosen to be l=2d. We can see that, after cutting the KC${}_{8}$-SL into nanoribbons, the Drude asymmetric tail transforms into a series of IR-active DPR n=1,3,5,7,… with the energy depending on the nanoribbon width *d*. As expected, as *d* increases, the energy of the DPR decreases and the energy difference between them becomes smaller. Considering that the sample is driven by the electromagnetic field, homogeneous in the *y* direction, the peaks appearing in the absorption spectra obviously represent optically active dipolar modes. According to the continuity equation ${\dot{\rho }}^{ind}=-\frac{\partial }{\partial y}j_{y}^{ind}$ and $j^{ind}=\sigma ⊗E$, where E is the external field given by Equation (Equation 1), the induced density can be calculated from $\rho^{ind}∼ℜ\partial \sigma_{yy}/\partial y$. Indeed, Figure 5a, which shows the induced electronic densities at frequencies ω corresponding to the absorption peaks n=1,3,5,7 in Figure 4a, clearly demonstrates their dipolar character.

On the other hand, the induced densities of the other non-dipolar modes n=2,4,6,… are zero, so they represents the dark modes, not visible in the absorption spectrum. For the period chosen here (l=2d), the separation between the nanoribbons *d* is quite large, and the interaction between the dipolar modes in the different nanoribbons is negligible; thus, they can be considered as almost decoupled resonances of the individual nanoribbons. However, we show below that these modes are still weakly dispersive as we increase the wave vector $Q_{y}$, suggesting their small but finite interaction.

Figure 6 shows the absorption intensities A for different wave vectors $Q_{y}$ (mostly outside the radiative region $Q_{y}>\omega /c$) in the KC${}_{8}$-NR arrays of widths: (a) d=50 nm; (b) d=100 nm; (c) d=200 nm. The figure also compares them with the absorption intensities in the KC${}_{8}$-SL shown in Figure 6d. The nanoribbon period is again l=2d. The red doted lines denote the energies of the IR active modes (n=1,3,5,…), as they appear in Figure 4, while the turquoise doted lines denote the energies of the dark modes (n=2,4,6,…). Green dotted lines denote the dispersion relation of the Dirac plasmon in the KC${}_{8}$-SL. We can see that the principal mode n=1 is the most dispersive one (within each BZ) and the most intensive in the first two BZ $0<Q_{y}<4\pi /l$. The rest of the modes, n=2,3,4,5,…, are less dispersive (flat patterns within some of the exBZs), and they are the most intensive through the few extended BZ, but precisely in the way that resembles the DP dispersion relation. This is particularly noticeable for the larger widths *d*, as can be seen by comparing Figure 6c,d. Moreover, we can see that the modes n=3,5,… in the extended BZ are ‘folded’ (although with much lower intensity) into the radiative region ($Q_{y}\approx 0$) where they become IR active. This is clearly noticeable, e.g., for the n=3 mode in Figure 6a–c, where just a small fraction of that mode, in exBZ, ‘projects’ into the radiative region ($Q_{y}=0$). This is in accordance with the results in Figure 4, where just the principal mode n=1 is the most intensive, while the higher modes n=3,5,7… are significantly suppressed.

These results show us that we can fold the fragments of the Dirac plasmon (at $Q_{y}\approx 2\pi \left(2k+1\right)/l;k=0,1,2,\ldots$) into the radiative region and make them IR-active resonances, in a controlled way, by cutting the KC${}_{8}$-SL into an array of nanoribbons, as sketched in Figure 5b,c. For example, by changing the nanoribbon width, we can choose which part of the Dirac plasmon in the KC${}_{8}$-SL we want to fold into the radiative region. It should be noted that this procedure is also valid in the opposite direction, starting from a single nanoribbon antenna. A single nanoribbon supports nondispersive and localized (both IR active and dark) plasmon resonances, but, after the nanoribbons are arranged in a lattice, the plasmon resonances unfold over the extended BZ resembling the DP, regardless of the period *l*. All these manipulations are experimentally feasible, which could have a significant impact on the applied plasmonics.

### DPR Band Structure

We now analyze the dispersion relations of the DPR within the 1stBZ, $Q_{y}\in \left[-\pi /l,\pi /l\right]$, i.e., the DPR band structure. In the previous examples, the separation between the nanoribbons is quite large, so the coupling between the DPR in the different ribbons is weak and, consequently, the DPR are weakly dispersive within the 1stBZ. However, we show above that the dispersion across the exBZs is strong so that it resembles the Dirac plasmon in the KC${}_{8}$-SL. The origin of this dispersivity is simple: for larger wave vector $Q_{y}$, the spatial variation of the external field partially or fully fits the spatial variation (or symmetry) of higher excited modes n=2,3,4…, regardless of whether they are dipolar or non-dipolar modes, and finally it efficiently excites these modes. On the other hand, the homogenous external field ($Q_{y}=0$) selectively and quite inefficiently excites the higher dipolar modes n=3,5,7,… Therefore, the inter-zonal dispersivity always exists, even though the interaction between the ribbons is negligible. However, for smaller separations, the interaction between the nanoribbons is getting stronger and DPRs become dispersive within the 1stBZ (or within some individual exBZ).

Figure 7 shows the absorption intensities A in the KC${}_{8}$-NR array deposited on the Al${}_{2}$O${}_{3}$ surface for different wave vectors $Q_{y}\in$1stBZ. The nanoribbon width is d=200 nm and the period is l=220 nm. The photon dispersion ω=Qc is also shown (green dotted lines) in order to denote the radiative region ω>Qc. It can be noticed that this small separation (l−d=20 nm) causes substantial dispersivity of the principal n=1 mode, resulting with the band width of about W=80 meV and the band gap opening of about $E_{g}=60$ meV. The higher DPR (n=2,3,4…) are less dispersive, probably because they produce short-ranged electric field so the inter-ribbon interaction is weaker. It is interesting that the dark mode n=2 seems to split into two branches as $Q_{y}$ decreases. This may be the evidence of the surface or ‘edge’ plasmons localized at the KC${}_{8}$-NR boundaries [17]. The edge plasmons are the counterparts of the standard surface plasmons appearing on metal surfaces. They are the extra solutions of the Maxwell’s equation, due to the symmetry breaking caused by the edge, and have an evanescent character, in contrast to the DPR which oscillate across the nanoribbon (see Figure 5a). The DPR band-gap can be manipulated by changing the KC${}_{8}$-NR parameters which opens the possibility for trapping the photons in the principal n=1 band and achieving the Dirac plasmon Bose–Einstein condensate. Therefore, the doped graphene nanoribbons enable direct light–plasmon interaction, which can be exploited in many plasmonic or photonics applications, while at the same time it can serve as a polygon for exploring fundamental physical phenomena such as strong light–matter interactions, which has been intensively explored recently [41].

## 5. Comparison with Experiment

In order to verify the accuracy and experimental feasibility of the above results, we compare them with some recent experimental results. Since optical absorption experiments on the KC${}_{8}$-NR arrays still do not exist, we compare our results with the experimental results for the differential transmission $\Delta T=T-T_{CNP}$ through the array of doped graphene micro-ribbons on Si/SiO${}_{2}$ substrate [15], where $T_{CNP}$ is the transmission coefficient through the device at the charge neutral point (CNP). ΔT is directly related to our infrared absorption spectrum A. In our calculations, the graphene is doped by holes, where the hole concentration is chosen to be $1.5\times 10^{13}$ cm${}^{-2}$ [15] ($E_{F}=-0.374$ eV with respect to the Dirac point). The Si/SiO${}_{2}$ substrate is described by the dielectric constant $\epsilon_{s}=6.5$, which is between 3.8 in SiO${}_{2}$ and 11.7 in Si. The separation between the graphene and the SiO${}_{2}$ surface is taken to be $z_{0}=4$ Å [42].

In order to gain insight into the measured data for wider energy range, in Figure 8a, we compare the experimental result for ΔT (blue circles) with our results for A calculated for two intraband damping parameters $\eta_{intra}=1$ meV (brown line) and $\eta_{intra}=15$ meV (turquoise line) and on extended frequency scale. The graphene ribbon width is d=1μm and the period is l=2μm. We can see that for the smaller damping parameters absorption spectra shows DPR n=1,3,5,…, which for the larger damping smooth out into an asymmetrical lineshape which is in excellently agreement with the experimental data. Therefore, we can conclude that the experimental lineshape mainly consists of the principal dipolar mode n=1, and its asymmetry is a consequence of the excitations of higher-order dipolar modes n=3,5,… Figure 8b shows the theoretical absorption spectra A in an array of graphene ribbons of widths d=1μm (blue line), 2 μm (red line) and 4 μm (green line) and compares them with the experimental results for ΔT (blue, red and green circles). The period is l=2d. These results undoubtedly confirm that the broad experimental peaks corresponds to the n=1 DPR, while the higher-order dipolar DPR n=3,5,7,… give the spectrum an asymmetric shape. Moreover, these results determine the natural intraband damping parameter $\eta_{intra}=15$ meV, which is a consequence of the electron–phonon (instrinsic and SO phonons) interactions, scattering on impurities and other crystal imperfections.

All this suggests that the electron–phonon interaction (or maybe some other scattering mechanisms) is likely to play a significant role in profiling the higher-order plasmon resonances *n* = 3, 5, …. Below, we show that the synthesis of the potassium intercalated graphene (KC${}_{8}$) ribbons is indeed possible and explore how the potassium adatoms influence the strength of the electron–phonon coupling. The latter is very important because alkali metals can sometimes increase and sometimes decrease the strength of the electron coupling to the graphene E${}_{2g}$ phonon [43], which is, as already mentioned, very important for the damping of the plasmon resonances.

Our scanning electron microscopy (SEM) analysis of GNRs placed on a carbon tape and analyzed with 5 kV Helios NanoLab DualBeam scanning electron microscope showed multilayer GNR structures with lengths of several microns and widths ranging around 100 nm. Figure 9a confirms a flat-rippled multilayer nanoribbon morphology. These GNRs were synthesized via CVD following the procedure described in [44] and further intercalated by conducting a two-zone vapor transport method, as described in [45]. The intercalation compounds was obtained by the combination of an alkali metal (K) placed in a glass vial with GNRs sealed under high vacuum conditions at 10${}^{-6}$ mbar in a proportion of 3.2 mg of GNRs per 1.3 mg of potassium (∼KC${}_{8}$ GNRIC). The characteristic Raman spectrum from the GNRs (Figure 9b–i) revealed the characteristic D-band and G-band located at ∼1338 cm${}^{-1}$ and ∼1574 cm${}^{-1}$, respectively. The D-band exhibited a larger intensity caused by the edge ripple proportion, while the D/G ratio was found to be 1.25 characteristic of graphene nanoribbons [44]. At ∼2674 cm${}^{-1}$, we observed the 2D-line characteristic of graphitic GNRs [44,46]. An intercalation process using the discussed pristine sample was performed obtaining a KC${}_{8}$ GNRIC. This sample was kept under vacuum conditions to avoid oxidation during the Raman measurements. The Raman spectrum obtained from the KC${}_{8}$ GNRIC in Figure 9c shows the characteristic broad Fano-line-shape composed by a G-line at ∼1505 cm${}^{-1}$ caused by the intercalation of potassium layers in between the graphene ribbons. This characteristic G-line in intercalation compounds originates from strong electron–phonon coupling (EPC) interactions existing between the potassium atoms and the graphene layers, as we reported previously for graphite intercalation compounds [47]. It is proven that by doping graphene with electron donors the Dirac plasmon resonances can be excited [18]. Thus, here we could introduce the fact that, by obtaining a highly e${}^{-}$-doped intercalation compound (i.e., KC${}_{8}$), we must obtain: (i) a strong EPC that will be responsible for superconductivity in stage I GICs according to the BCS theory directly related to the G-line phonon frequency and to the adiabatic ($\omega_{A}$) and non-adiabatic ($\omega_{NA}$) phonon frequencies of the GIC; (ii) a strong EPC in GNRIC will serve to excite plasmonic resonances derived from the perpendicular electric field in those nanostructures. To estimate the renormalized electron–phonon scattering line width ($\Gamma_{EPC}$) in GNRIC, we consider the G-line phonon frequency ($\omega_{G}$ from $E_{2g}$) of the Raman spectrum in Figure 9c in the following equation [47]:(27)$$\frac{\gamma^{EPC}}{2}=\sqrt{\left(\omega_{G}-\omega_{A}\right)\left(\omega_{NA}-\omega_{G}\right)}$$
where $\omega_{G}$ is the measured G-line frequency from the Fano function (1505 cm${}^{-1}$), $\omega_{A}$ is the adiabatic phonon frequency (1223 cm${}^{-1}$) and $\omega_{NA}$ corresponds to the non-adiabatic phonon frequency (1534 cm${}^{-1}$) [48]. From this equation we obtained, $\Gamma_{EPC}=$243 cm${}^{-1}$ for KC8 GNRIC is indicative of a potential superconducting behavior of the material as it behaves linearly with the measured FWHM ($\Gamma_{ph}=$262 cm${}^{-1}$). An individualized GNRIC can be evinced in Figure 9d to confirm no further damage to the structure of the graphitic ribbon.

Finally, our theoretical results are in excellent agreement with the experimental results, confirming the credibility of the presented method and the above-stated conclusions.

## 6. Conclusions

We developed the ab initio theoretical formulation of the electromagnetic response in doped graphene nanoribbons and used it to calculate the optical absorption in an array of potassium-doped graphene nanoribbons deposited on a dielectric Al${}_{2}$O${}_{3}$ surface. We demonstrated that the replacement of the single-layer doped graphene by the graphene ribbons of different period *l* causes the ‘projection’ of the Dirac plasmon into the radiative region, turning it into a series of IR-active Dirac plasmon resonances. This encourages the fabrication of graphene nanoribbons with the desired electromagnetic response in the IR or THz frequency range, which could be used in plasmonic, photonic or optoelectronic applications. We showed that the DPR band structure (band gap $E_{g}$ and band width *W*) can be tuned by changing the period *l*. By creating a large band gap $E_{g}$, one can enable trapping of the photons in the principal n=1 band and achieve the Dirac plasmon Bose–Einstein condensate. Therefore, the graphene ribbons can be exploited in many plasmonic or photonic applications, but at the same time they can serve as a polygon for exploring fundamental physical phenomena such as strong light–matter interactions.

## Figures and Tables

**Figure 1 materials-14-04256-f001:**
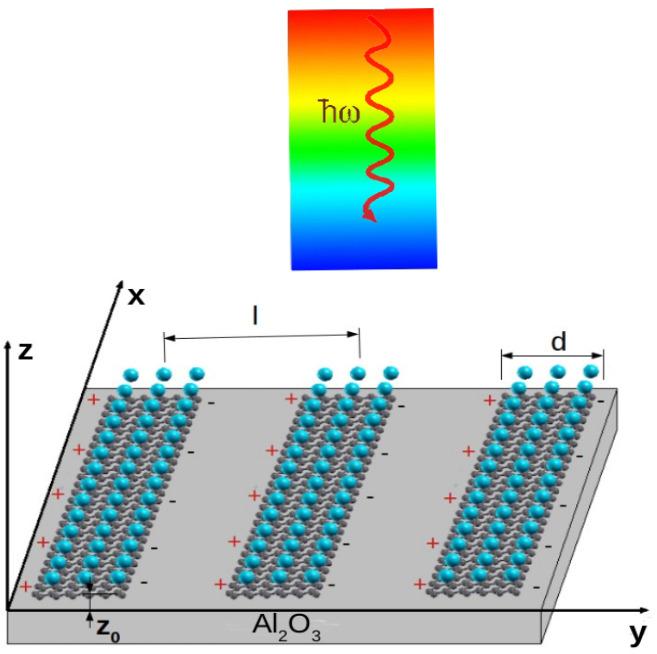
Array of KC${}_{8}$-NR of width *d* and period *l* deposited on a dielectric Al${}_{2}$O${}_{3}$ surface. The separation between the graphene plane and the dielectric surface is $z_{0}$.

**Figure 2 materials-14-04256-f002:**
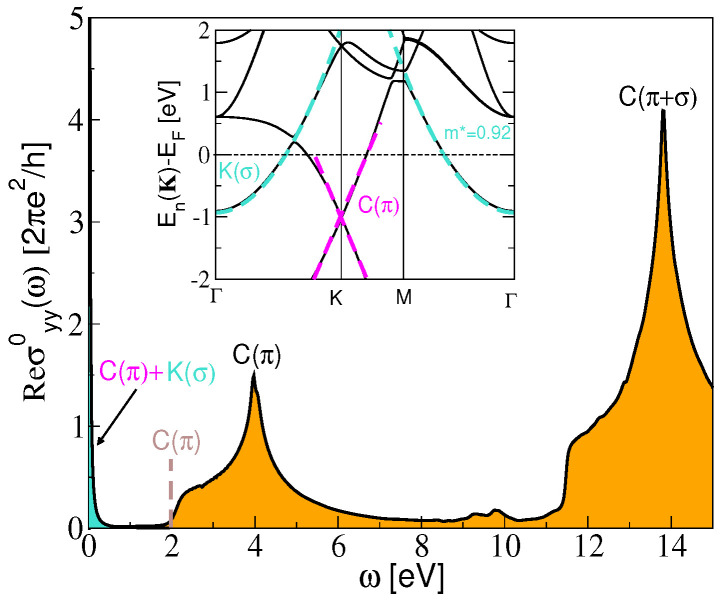
The ab initio optical conductivity $ℜ\sigma_{yy}^{0}$ in the KC${}_{8}$-SL. The intraband $\sigma^{\mathrm{intra}}$ and the interband $\sigma^{\mathrm{inter}}$ contributions are turquoise and orange shaded, respectively. The insert shows the KC${}_{8}$-SL band structure. Turquoise dashed lines show the parabolic fit of the K(σ) band for the effective mass $m^{*}=0.92$. Magenta dashed lines denote the graphene cone.

**Figure 3 materials-14-04256-f003:**
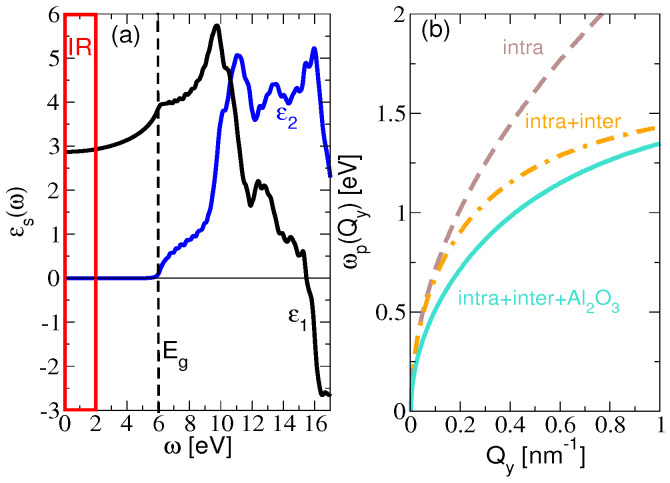
(**a**) *Ab initio* macroscopic dielectric function $\epsilon_{s}\left(\omega \right)=\epsilon_{1}\left(\omega \right)+i\epsilon_{2}\left(\omega \right)$ of the bulk Al${}_{2}$O${}_{3}$ crystal. Red frame denotes the frequency range of the IR active plasmons studied here; (**b**) Dispersion relations of the Dirac plasmon in the KC${}_{8}$-SL obtained from the maxima of the real part of the screened conductivity (Equation 26), where $\sigma^{0}=\sigma^{\mathrm{intra}}$ and $\Gamma =\Gamma^{0}$ (brown dashed), $\sigma^{0}=\sigma^{\mathrm{intra}}+\sigma^{\mathrm{inter}}$ and $\Gamma =\Gamma^{0}$ (orange dashed-dotted) and $\sigma^{0}=\sigma^{\mathrm{intra}}+\sigma^{\mathrm{inter}}$ and $\Gamma =\Gamma^{0}+\Gamma^{sc}$ (solid turquoise).

**Figure 4 materials-14-04256-f004:**
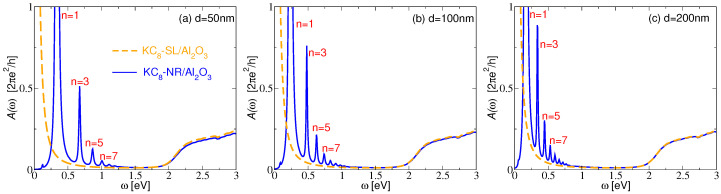
The absorption spectra of the KC${}_{8}$-NR of thicknesses (**a**) d = 50 nm, (**b**) d = 100 nm and (**c**) d = 200 nm (blue lines) and the absorption spectra of the KC${}_{8}$-SL (orange dashed lines). Both structures are deposited on a dielectric model Al${}_{2}$O${}_{3}$ surface. The nanoribbon period is l=2d and $z_{0}=3.0$Å.

**Figure 5 materials-14-04256-f005:**
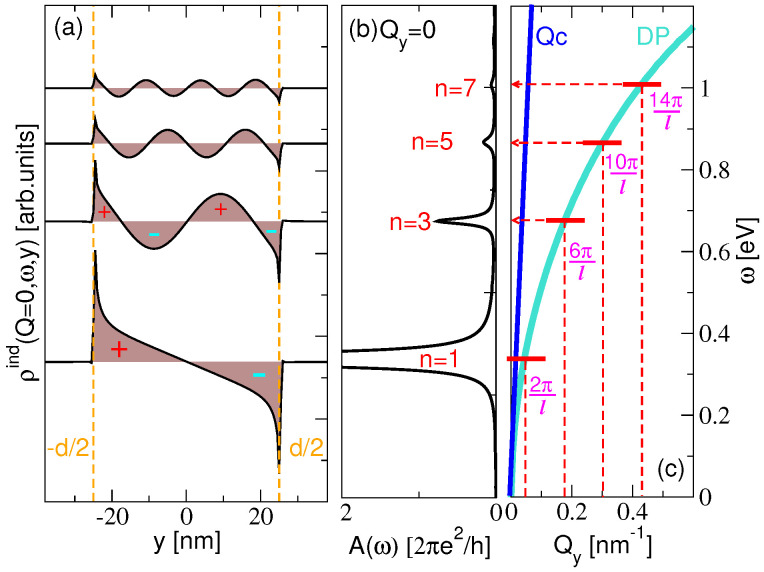
(**a**) The induced electronic densities ($\rho^{ind}∼ℜ\partial \sigma_{yy}/\partial y$) calculated for $Q_{y}=0$, at the frequencies corresponding with the IR active resonances n=1,3,5,7, in the KC${}_{8}$-NR of widths d=50 nm deposited on a dielectric model Al${}_{2}$O${}_{3}$ surface. The nanoribbon period is l=2d and $z_{0}=3.0$ Å; (**b**) The absorption spectrum corresponding to the modes in (**a**); (**c**) Schematic presentation of the ‘projection’ of the fractions of the DP (at $Q_{y}\approx 2\pi \left(2k+1\right)/l;k=0,1,2,\ldots$) into the radiative region where they become IR active resonances.

**Figure 6 materials-14-04256-f006:**
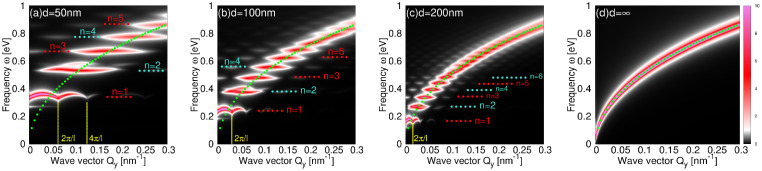
(**a**–**c**) The absorption intensity A (in units $2\pi e^{2}/h$) in the KC${}_{8}$-NR arrays of widths: (**a**) d = 50 nm; (**b**) d = 100 nm; (**c**) d = 200 nm. (**d**) The absorption intensity A in the KC${}_{8}$-SL. Red dotted lines denote the energies of the IR active modes n=1,3,5,…. Turquoise dotted lines denote the energies of the dark modes n=2,4,6…. Structures are deposited on a dielectric model Al${}_{2}$O${}_{3}$ surface. The nanoribbon period is l=2d and $z_{0}=3.0$ Å. Green dotted lines represent the DP dispersion in KC${}_{8}$-SL.

**Figure 7 materials-14-04256-f007:**
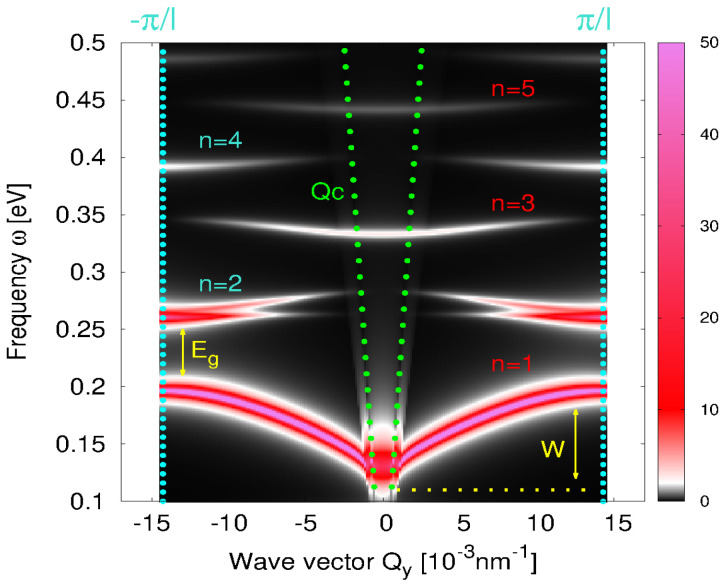
The absorption intensities A (in units $2\pi e^{2}/h$) in an array of KC${}_{8}$-NR deposited on a dielectric model Al${}_{2}$O${}_{3}$ surface for $Q_{y}\in \left[-\pi /l,\pi /l\right]$ showing the DPR band structure. The nanoribbons width is d=200 nm, the period is l=220 nm and $z_{0}=3.0$ Å. Green doted lines denote the photon dispersion relation ω=Qc. Turquoise dotted lines denote the 1stBZ boundaries.

**Figure 8 materials-14-04256-f008:**
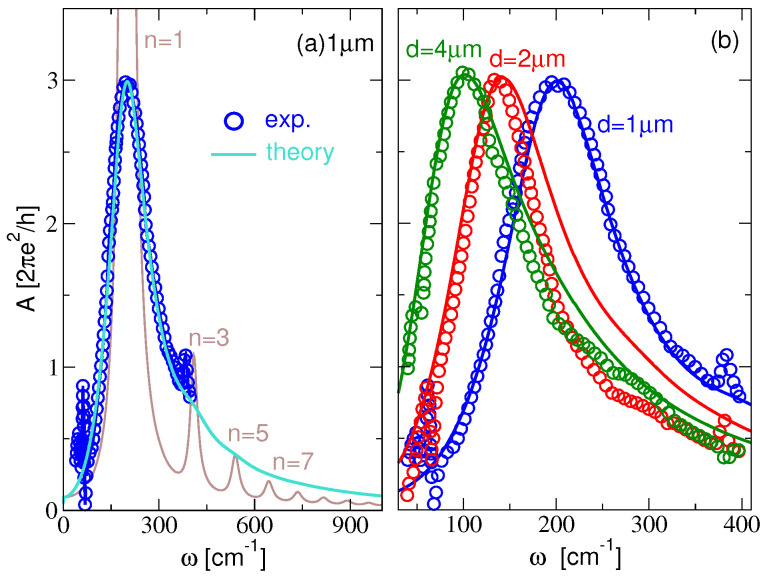
(**a**) Comparison of the absorption spectra A in a doped graphene ribbons array of width d=1μm calculated for $\eta_{intra}=1$ meV (brown line) and for $\eta_{intra}=15$ meV (turquoise line) with the result of differential transmission ΔT [15] (blue circles); (**b**) Comparison of the absorption spectra A in doped graphene ribbons array of widths d=1μm (blue line), d=2μm (red line) and d=4μm (green line) with the result of the differential transmission ΔT [15] (blue, red and green circles). Here, the intra-band damping parameter is $\eta_{intra}=15$ meV, the period l=2d and $z_{0}=4$ Å in all cases and the hole concentration is $1.5\times 10^{13}$ cm${}^{-2}$ ($E_{F}=-0.374$ eV relative to Dirac point).

**Figure 9 materials-14-04256-f009:**
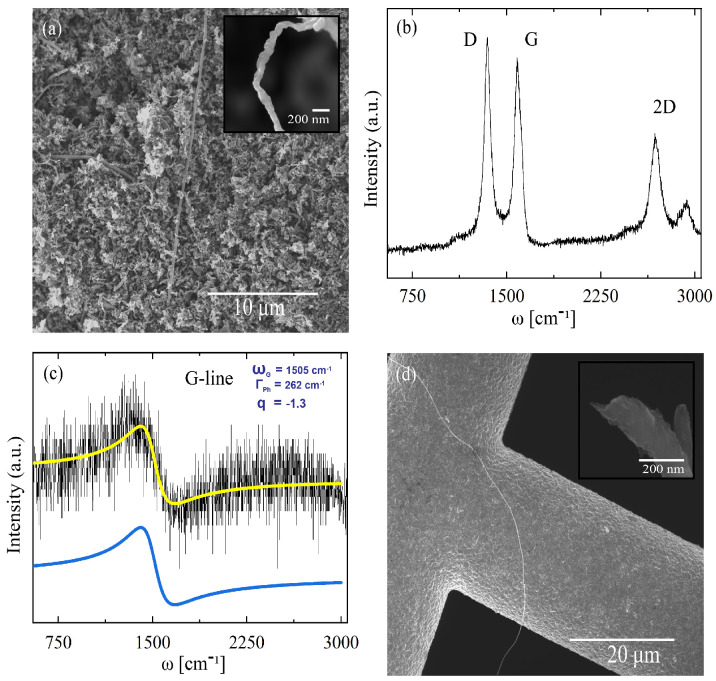
(**a**) SEM micrograph from pristine GNRs as obtained from the CVD process. A wide bundle of ribbons are evinced in widths of around 100 nm and lengths higher than 10 microns; (**b**) Raman spectra of pristine graphene nanoribbons. Characteristic GNRs features are present. The double resonance single 2D-line component around 2700 cm${}^{-1}$ indicates the presence of a graphene-like ribbon structure; (**c**) Raman spectrum of graphene nanoribbon intercalation compounds (GNRIC) in a KC${}_{8}$ stoichiometry. The Fano-like line shape derived from the strong coupling between K atoms and the graphene is a fine characteristic of a stage I intercalation compound; (**d**) Individualized GNRs after potassium intercalation. The structure and shape of the GNRs was not affected after the intercalation as observed in the micrograph. GNRs were confirmed to be longer than 10–20 microns with a width ∼100–200 nm.

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
