# Peer review of "Infra-Red Active Dirac Plasmon Serie in Potassium Doped-Graphene (KC8) Nanoribbons Array on Al2O3 Substrate"

_materials, 2021, doi:10.3390/ma14154256_

Round 1

Reviewer 1 Report

In this manuscript, the authors proposed a theoretical formalism to study the plasmonics in an array of potassium doped graphene nanoribbons (KC8) on a dielectric substrate. They performed first-principles calculations and demonstrated the decomposition of Drude tail in a single layer KC8 into Dirac plasmon resonances in periodic KC8 ribbons. Comparison between theory and differential transmission experiments shows good agreement. The results presented in the manuscript are interesting and comprehensive. However, the following questions have to be addressed:

  1. Conductivity, as an intensive physical quantity, is ill-defined in 2D materials. It depends on the thickness of the supercell used in the first-principles calculations. For example, when the vacuum layer is chosen to be infinitely thick, then the as-calculated conductivity would vanish. In order to compare results calculated with different supercell thickness, I think the authors should come up with a rescaling scheme for the conductivity.

  1. The authors only considered homogeneous broadening (Eqs. 17 and 19) in the expression of conductivity, which has a Lorentzian profile. However, in real experiments, there will likely be strong inhomogeneous broadening in a Gaussian profile. Would additional inhomogeneous broadening improve the agreement between theory and experiment in FIG. 8(b)?

  1. In Eq. (7), the authors only considered Gvectors along the y-axis, but the system is actually 2D instead 1D. Would the inclusion of Gvectors along the x-axis modify the results for small (l - d)?

  1. In Eq. (17), which quantity on the right-hand side is Q-dependent?

  1. Al2O3 is hexagonal and therefore the dielectric constants along the c-axis and perpendicular to the c-axsi are different. In Eq. (24), the authors only consider the longitudinal dielectric response function, which may not correspond to the true dielectric constant experienced by the external transverse electromagnetic field.

  1. I cannot find computational details for the KC8-NR structures. Were KC8-NR treated with DFT or some other modeling techniques? If it was calculated with DFT, were the edges of graphene nanoribbons passivated by hydrogen?

  1. In Sec. III, the authors mentioned that “the onsite for the interband transitions between the graphene C(π) bands appear at 2eV”, but I can hardly find the corresponding interband transition in FIG. 2. Please explain.

  1. How to understand that the induced density is proportional to ∂σyy/∂y?

  1. There are some typos in referring to equations in Sec. II A: “Assuming that the screened conductivity σ can be transformed the same way as the nonlocal irreducible polarizability 7, and using the expansions (7) and (9), …”

  1. In Sec. II C & D, the letters to denote the momentum transfer are inconsistent. In Sec. II C, it is Q, while in Sec. II D, q is used. So do k and K. Please make sure that the letters are used consistently.

  1. The authors used the abbreviation of “KC8-GNR” in the abstract but later on they switched to “KC8-NR”.

Author Response

Refree:

In this manuscript, the authors proposed a theoretical formalism to study the plasmonics in an array of potassium doped graphene nanoribbons (KC8) on a dielectric substrate. They performed first-principles calculations and demonstrated the decomposition of Drude tail in a single layer KC8 into Dirac plasmon resonances in periodic KC8 ribbons. Comparison between theory and differential transmission experiments shows good agreement. The results presented in the manuscript are interesting and comprehensive. However, the following questions have to be addressed:

Authors response:

We thank the referee for the positive review, for the detailed reading of the manuscript and for pointing out the omissions. We corrected the manuscript in accordance with the referee’s suggestions as much as possible.

Refree:

  1. Conductivity, as an intensive physical quantity, is ill-defined in 2D materials. It depends on the thickness of the supercell used in the first-principles calculations. For example, when the vacuum layer is chosen to be infinitely thick, then the as-calculated conductivity would vanish. In order to compare results calculated with different supercell thickness, I think the authors should come up with a rescaling scheme for the conductivity.

Authors response:

We thank the referee for this important remark about the conductivity tensor. We should emphasize that our KC8-SL conductivity does not depend on the vacuum thickness L. Namely we calculate nonlocal spatial dependent conductivity as:

\[

\sigma(z,z’)=\frac{1}{L}\sum_{G_zG_z’}\sigma_{G_zG_z’}e^{iG_zz-iG_z’z’}

\]

which is independent on the vacuum thickness L. Only the number of the Fourier components needed to expand it depends on L. Finally, we integrate that nonlocal function over z and z’ and get

\[

\sigma(\omega)=L\sigma_{G_z=0G_z’=0}

\]

which is the KC8-SL conductivity presented in the Sec.IIC.

We added small paragraph to explain this at the beginning of the Sec.IIC.

Referee:

  1. The authors only considered homogeneous broadening (Eqs. 17 and 19) in the expression of conductivity, which has a Lorentzian profile. However, in real experiments, there will likely be strong inhomogeneous broadening in a Gaussian profile. Would additional nhomogeneous broadening improve the agreement between theory and experiment in FIG. 8(b)?

 Authors response:

This remark is indeed interesting and relevant. We noticed that the agreement with the experiment significantly depends on the homogenous broadening we use. We even think that we can extract the damping constant by fitting our results to the experimental data as demonstrated in Fig8a. Frequency dependent broadening would probably increase the agreement with the experiments, especially if the plasmon resonance enters the region of phonon frequencies. In order to activate the phonon effect, frequency-dependent damping could be calculated by using e.g. a Memory function approach, which requires the ab initio calculation of the electron phonon coupling functions, which goes beyond the scope of our research. However, we intend to include this effect it in the future.

Referee:

  1. In Eq. (7), the authors only considered Gvectors along the y-axis, but the system is actually 2D instead 1D. Would the inclusion of Gvectors along the x-axis modify the results for small (l - d)?

 Authors response:

The reciprocal wave vectors G contribute only in the direction in which the translational invariance is broken. Considering that we chose that the nanoribbons are parallel to the x axis, the symmetry is broken in the y direction so the wave vector G_y naturally appears in that direction. We could maybe also study the electromagnetic modes propagating in the Q_x direction, but such modes would not be optically active so we do not analyze them in order to avoid confusion. Finally, the effect of the crystal wave vectors (G_x, G_y) (so called crystal local field effects) would be important only for the microscopic d and l (comparable to the crystal unit cell) and for the large non-optical wave vectors Q, which is beyond the scope of this investigation.

Refree:

  1. In Eq. (17), which quantity on the right-hand side is Q-dependent?

  Authors response:

We thank the referee for pointing out this omission. We removed Q in Eq.17.

Refree:

  1. Al2O3 is hexagonal and therefore the dielectric constants along the c-axis and perpendicular to the c-axsi are different. In Eq. (24), the authors only consider the longitudinal dielectric response function, which may not correspond to the true dielectric constant experienced by the external transverse electromagnetic field.

   Authors response:

The referee is right, we described the dielectric response of Al2O3 by using the macroscopic longitudinal dielectric function \epsilon_M. However, it can also be considered as some averaged dielectric function. Namely, \epsilon_M includes both longitudinal and transversal electronic excitations with a field which is in the end averaged, so that it appears as an isotropic tensor. Therefore, we can say that the transversal response is partially included in our calculation. However, as the referee noted, for more accurate inclusion of the anisotropic substrate’s polarization, its dielectric function should be divided into \epsilon_\perp (response along c-axis) and \epsilon_\parallel (response along a-axis). In this paper we did not consider this option but we will take it into account in our future calculations. We thank the referee for pointing this out.

Refree:

  1. I cannot find computational details for the KC8-NR structures. Were KC8-NR treated with DFT or some other modeling techniques? If it was calculated with DFT, were the edges of graphene nanoribbons passivated by hydrogen?

   Authors response:

The electromagnetic response of the KC8-NR structure is described by the model local conductivity defined by Eqs.5-8. The only input calculated with DFT is the KC8-SL conductivity, and the calculation is described in Sec.IIC. We believe that this modeling is fully justified for the sub-micrometer or micrometer ribbons and in the optical limit where \lamda>>>a, which is the limit we investigated here. However, the dielectric response of the few nanometer ribbons becomes strongly nonlocal (due to the neighboring edges) and must be calculated entirely from the first principles including the hydrogen passivation, as it was done in Ref.[20]. In our case the nanoribbons thickness is up to 4 micrometers, so we are far beyond this full ab initio limit.

Refree:

  1. In Sec. III, the authors mentioned that “the onsite for the interband transitions between the graphene C(π) bands appear at 2eV”, but I can hardly find the corresponding interband transition in FIG. 2. Please explain.

 Authors response:

The onset for the interband C(π) transition can be recognized as the ‘step-like’ function appearing at about \omega=2eV in the KC8 interband conductivity. We denoted it by a dashed vertical line in new version of Fig.2. and added the appropriate text.

Refree:

  1. How to understand that the induced density is proportional to ∂σyy/∂y?

 Authors response:

The induced density \rho^{ind} and the induced current j^{ind} are related by the continuity equation as

\[

\dot{\rho^{ind}}=-\frac{\partial j^{ind}_y}{\partial y}.

\]

The induced current is driven by the electric field, given by Eq.1, so that

\[

j^{ind}\sim \sigma(y,g’=0) (A)

\]

and

\[

\dot{\rho^{ind}}\sim \rho^{ind}(y) (B)

\]

Finally, combining (A) and (B) we obtain that \rho^{ind}(y)\sim\frac{\partial\sigma}{\partial y}.

We added the corresponding explanation in the new version of the manuscript.

Referee:

  1. There are some typos in referring to equations in Sec. II A: “Assuming that the screened conductivity σ can be transformed the same way as the nonlocal irreducible polarizability 7, and using the expansions (7) and (9), …”

Authors response:

We thank the referee for pointing out this omission. We corrected that sentence and now it reads:

Using the expansions (7) and (9), and assuming that the screened conductivity σ can be transformed the same way as the conductivity σ0 (expansion Eq.7), the Dysons equation (4) transforms into matrix equation for the screened conductivity’

Referee:

  1. In Sec. II C & D, the letters to denote the momentum transfer are inconsistent. In Sec. II C, it is Q, while in Sec. II D, q is used. So do k and K. Please make sure that the letters are used consistently.

 Authors response:

The capital letter K and Q refer to the 2D wave vector and transfer wave vector (appearing in the conductivity or in the propagator \Gamma) while the lowercase letters k and q refer to the 3D wave vector and transfer wave vector (appearing in the Al2O3 macroscopic dielectric function). This is clearly stated throughout the text.

Referee:

  1. The authors used the abbreviation of “KC8-GNR” in the abstract but later on they switched to “KC8-NR”.

Authors response:

Again, we thank the referee for pointing out this omission. We corrected it by replacing KC8-GNR with KC8-NR in the abstract.

Reviewer 2 Report

The array of doped graphene nanorib-bons deposited on a dielectric substrates is important for optoelectronics and spintronics,especially when  combined with  localized plasmons. This study developed the ab initio theoretical formulation of the electromagnetic response in the doped graphene nanoribbons and applied it to the optical absorption in  potassium doped graphene nanoribbons array  on  Al2O3 surface. The ’projection’ of the Dirac plasmon was observed as  a series of IR active Dirac
plasmon resonances due to the projection effects, which means the possible design for the desired electromagnetic response in the IR or THz frequency range, which could be used in plasmonic, photonic or optoelectronic
, and even  fundamental physical phenomena such as strong
light-matter interaction.  I believe this works is interesting. And I would like to see its publications.

Author Response

Referee:

The array of doped graphene nanoribbons deposited on a dielectric substrates is important for optoelectronics and spintronics, especially when combined with localized plasmons. This study developed the ab initio theoretical formulation of the electromagnetic response in the doped graphene nanoribbons and applied it to the optical absorption in potassium doped graphene nanoribbons array on Al2O3 surface. The ’projection’ of the Dirac plasmon was observed as a series of IR active Dirac plasmon resonances due to the projection effects, which means the possible design for the desired electromagnetic response in the IR or THz frequency range, which could be used in plasmonic, photonic or optoelectronic, and even fundamental physical phenomena such as strong light-matter interaction. I believe this works is interesting. And I would like to see its publications.

Authors response:

We thank the referee for the positive review, for the detailed reading of the manuscript and for pointing out the important contributions of this investigations.

Reviewer 3 Report

In the manuscript “Infra-red active Dirac plasmon serie in potassium doped-graphene (KC8) nanoribbons array on Al2O3 substrate” by Jakovac, et.al., the authors developed an ab initio theoretical formulation of the electromagnetic response in the doped graphene nanoribbons and used that to calculate the optical absorption in potassium doped graphene nanoribbons deposited on the dielectric Al2O3 surfaces. In this work, the theoretical results have been compared with experimental measurements of different transmission through the same sample. It has been shown that the theoretical results are in good agreement with the experimental results, which further confirm the validity of the proposed theory. In the current manuscript, the theory seems right, it can indeed predict micrometer graphene nanoribbons intercalation compound. The theoretical results look convincing and sufficient and I think this work is worth publishing in Materials. My only suggestion is that in section 2, the mathematical explanation should be clearer. The author should stress the key physics part and improve the readability of this work.  

Author Response

Refree:

In the manuscript “Infra-red active Dirac plasmon serie in potassium doped-graphene (KC8) nanoribbons array on Al2O3 substrate” by Jakovac, et.al., the authors developed an ab initio theoretical formulation of the electromagnetic response in the doped graphene nanoribbons and used that to calculate the optical absorption in potassium doped graphene nanoribbons deposited on the dielectric Al2O3 surfaces. In this work, the theoretical results have been compared with experimental measurements of different transmission through the same sample. It has been shown that the theoretical results are in good agreement with the experimental results, which further confirm the validity of the proposed theory. In the current manuscript, the theory seems right, it can indeed predict micrometer graphene nanoribbons intercalation compound. The theoretical results look convincing and sufficient and I think this work is worth publishing in Materials. My only suggestion is that in section 2, the mathematical explanation should be clearer. The author should stress the key physics part and improve the readability of this work.  

Authors response:

We thank the referee for the positive review, for the detailed reading of the manuscript and for pointing out the omissions. We have modified the manuscript in accordance of the other referee's suggestions, and we believe the presentations is now more clear, including
the mathematical explanations.

Reviewer 4 Report

In the manuscript “Infra-red active Dirac plasmon series in potassium doped-graphene (KC_8) nanoribbons array on Al_2O_3 substrate” by J. Jakovic, L. Marusic, D. Andrade-Guevara, et al. The results of the theoretical study at the ab initio level of the electromagnetic response of the intercalated graphene ribbons placed on a dielectric substrate are presented. It is demonstrated how the Dirac plasmon dispersion is split into a series of Dirac plasmon resonances in the regular arrangement of nanoribbons of different sizes. A detailed description of how the absorption spectrum is transformed in the infra-red energy interval by varying geometrical parameters is presented.

I am convinced that this work presents a significant contribution to the field and maybe published without major revisions. I suggest to authors revise several points in the manuscript:

-In the Li-doped graphene on the same substrate [NPJ 2D Materials and Applications 4, 19 (2020)] in addition to the strong Dirac plasmon, a weak acoustic plasmon was found. Why such a plasmon does not exist in the present system despite two energy bands crossing the Fermi level?

-In Equation (3) the subscript “mu” present on the right side only. May absorption “A” depend on “mu”? In this case, how it correlates with the summation on “mu” in Eq. (2)?

- I suggest to shift the definition of \rho from below Eq. (8) to below Eq. (5) after its first appearance.

-What does mean \pm in the superscript in line 2 below Eq. (15)? How do “+” and “-” enter on the right side of the expression?

-It is not clear what does mean \alpha_{0,\pm} in line 4 below Eq. (15).

-How can one relay e^0_q, e^{\pm}_q, e^{0,\pm}_s, and  e^{0,\pm}_p in the text of subsection II.B?

-Definition of SBZ and S.B.Z. in Eqs. (18) and (19)?

-B.Z. in Eq. (21)?

-In some places, one can find the definition “l=2a”, in others “l=2d”. Why?

-In Fig. 9 it is difficult to resolve the panel (a). Also, check the notation of the panels in the caption.

Author Response

Referee:

In the manuscript “Infra-red active Dirac plasmon series in potassium doped-graphene (KC_8) nanoribbons array on Al_2O_3 substrate” by J. Jakovic, L. Marusic, D. Andrade-Guevara, et al. The results of the theoretical study at the ab initio level of the electromagnetic response of the intercalated graphene ribbons placed on a dielectric substrate are presented. It is demonstrated how the Dirac plasmon dispersion is split into a series of Dirac plasmon resonances in the regular arrangement of nanoribbons of different sizes. A detailed description of how the absorption spectrum is transformed in the infra-red energy interval by varying geometrical parameters is presented. I am convinced that this work presents a significant contribution to the field and maybe published without major revisions. I suggest to authors revise several points in the manuscript:

Authors response:

We thank to referee for the positive review, for the detailed reading of the manuscript and for pointing out the omissions. We corrected the new version of the manuscript correspondingly.

Refree:

-In the Li-doped graphene on the same substrate [NPJ 2D Materials and Applications 4, 19 (2020)] in addition to the strong Dirac plasmon, a weak acoustic plasmon was found. Why such a plasmon does not exist in the present system despite two energy bands crossing the Fermi level?

Authors response:

We thank to referee for pointing out this relevant point and for correlating it with our previous paper. Considering that 5th referee had similar comment we shall respond in a similar way. Namely, in order to see two plasmons (Dirac and Acoustic) the spatial dispersion of the dynamical response or inclusion of the crystal local field effects in the perpendicular (z) direction is mandatory. Here the spatial dispersion of the KC8 conductivity in z direction is squeezed in one single layer of zero thicknesses so that the two intraband contributions C(pi) and K(sigma) are fused into a single Drude-lake term with a large effective number of charge carriers. This approximation neglects the Acoustic plasmon but is fully justified in the optical limit where the weight of the AP is zero. However, for larger wave vectors Q the perpendicular dispersivity of the dielectric response becomes very import. For example, in that case the C(pi) and K(sigma) behave as two spatially separated 2D plasmas coupled via Coulomb interaction to produce the Acoustic and Dirac plasmon, which we first studied in our paper Phys. Rev. B 95 201408 (2017). We hope we have responded satisfactorily to this referee’s comment.

Referee:

-In Equation (3) the subscript “mu” present on the right side only. May absorption “A” depend on “mu”? In this case, how it correlates with the summation on “mu” in Eq. (2)?

Authors response:

Actually, the absorption depends on the polarization of the incident electromagnetic field (Eq.1). So if polarization of the incident field is not bare (i.e. neither of the components ex ey and ez is equal to zero) then both summations \mu and \nu in Eq.2 are active and A cannot be assigned a bare index \mu. However, if incident field is bare (eg. only ex is finite) then bot summations vanish so that A depends only on the \sigma_xx and becomes dependent on the bare component \mu. However, in order to be consistent with general expression Eq.2 we omit writing it explicitly.

Referee:

- I suggest to shift the definition of \rho from below Eq. (8) to below Eq. (5) after its first appearance.

Authors response:

We shifted the definition of \rho below Eq.5 as the referee suggested.

Refree:

-What does mean \pm in the superscript in line 2 below Eq. (15)? How do “+” and “-” enter on the right side of the expression?

Authors response:

Eq.15 represents the propagator of the scattered electric field. More specifically, for planar symmetry, it represents the electric field produced by an external point dipole and reflected at the dielectric surface. Therefore “-” represents the polarization of the incident electric field while “+” represents the polarization of the reflected electric field. Both polarizations should be included in the propagator Eq.15. \Gamma^{0} represents the 'direct' electric field produced by the point dipole, so that the superscript '0' represents the spherical forward propagating field. We added an appropriate sentence in the revised version of the paper, below Eq.15.

Referee:

-It is not clear what does mean \alpha_{0,\pm} in line 4 below Eq. (15).

Authors response:

The \alpha_{0,\pm} are simple factors that define the polarization vectors of the s and p polarized electromagnetic wave in the bare and scattered propagators \Gamma^0 and \Gamma^{sc}, respectively. It is explained in more detail in the previous point and in the manuscript.

Referee:

-How can one relay e^0_q, e^{\pm}_q, e^{0,\pm}_s, and  e^{0,\pm}_p in the text of subsection II.B?

Authors response:

The calculation of the s and p polarized electric field propagators is described in details in Ref.23 and 24, so we think it does not make sense to present a detailed description of the calculations in this paper as well.

Referee:

-Definition of SBZ and S.B.Z. in Eqs. (18) and (19)?

Authors response:

We thank the referee for pointing out this omission. 1.SBZ is 1st surface Brillouin zone and S.B.Z. also should be SBZ. We corrected the mistake and defined this abbreviation after Eq. 18.

Referee:

-B.Z. in Eq. (21)?

Authors response:

Again, we thank the referee for on pointing this out. Instead of B.Z. it should be 1.BZ and it represents the 1st Brillouin zone. We corrected this mistake and defined this abbreviation after Eq. 21.

Referee:

-In some places, one can find the definition “l=2a”, in others “l=2d”. Why?

Authors response:

We thank the referee. The d represents the ribbon thickness so everywhere through the text must be l=2d not l=2a. We corrected this mistake throughout the paper.

Referee:

-In Fig. 9 it is difficult to resolve the panel (a). Also, check the notation of the panels in the caption.

Authors response:

We corrected the Fig.9 and we wrote the caption more clearly.

Reviewer 5 Report

This is a very interesting paper, which combines theoretical and experimental investigation of potassium doped graphene nanoribbons (GNRs) on a dielectric substrate. First, detailed ab initio calculations of the optical properties of a periodic array of such GNRs are presented for an Al2O3 substrate, which are followed by a comparison of the theory with experiments for electrostatically doped GNRs on a Si/SiO2 substrate that were reported in Ref.[13]. Finally, the authors present their experimental confirmation of the synthesis of GNRs intercalated with K atoms. The paper provides a host of new and valuable information on those systems, so I recommend its publication after the authors consider the following minor comments.

  1. I think that the presentation of the paper needs more cohesion between its three major parts: (1) ab initio modeling of KC8-NRs on Al2O3, (2) comparison with experiment for electrostatically doped GNR on Si/SiO2 and (3) their own experimental Raman spectra of GNRIC. Namely, while the three parts share much in common, they nevertheless deal with somewhat different systems and/or different processes. I give some examples where a better connection between different parts is desirable.

(a) The doping mechanisms in parts (1) and (2) are different. The electrostatic doping in part (2) may give rise to a rather nonhomogeneous distribution of charge carriers across nanoribbons, whereas the doping by electron donation from the K-atoms should give a more homogeneous distribution. Perhaps the authors can comment on how homogeneous the K-atom density is expected to be across graphene in realistic nanoribbon structures.

(b) It is not clear what is achieved by the analysis in part (3) that would be relevant for parts (1) and (2). While part (3) discusses electron-phonon interaction in quite some detail, this interaction is neglected in part (1) and is only briefly mentioned as a possible mechanism that can give rise to a substantial enough broadening of the absorption spectra, which was found to be essential for achieving a good comparison with experiments [13] in part (2).

  1. I also have a few comments on the choice of modeling parameters. The authors approximate the KC8 as single layer of zero thickness, which is justified in the given range of wavelengths. In other words, they take the distance between graphene and the K-atom layer to be zero. In that respect, I have two questions.

(a) As a result of the above approximation, the two intraband contributions to the conductivity of KC8-SL seem to be fused into a single, strong Drude-lake term with a very large effective number of charge carriers. In view of their modeling of the K(sigma) and C(pi) bands in the inset of Fig.2, is it possible to estimate relative contributions of those two bands in the strong Drude peak?

(b) The authors keep the distance z0 between the KC8 and the substrate finite (3 or 4 Angstroms). It seems to me that taking z0 = 0 would be consistent with the zero-thickness approximation for the KC8-SL in the given range of wavelengths. The distance between graphene layer and the K-atom layer is on the order of 3 Angstroms, so if that is neglected, why not taking z0 = 0?

  1. My final comment refers to the very interesting remark on the observed splitting of the n=2 dark plasmon mode in the ab initio spectra in Fig.7. The authors state that “This may be the evidence of the surface or edge plasmons localized at the KC8-NR boundaries [15].” I am not entirely convinced that the electric field propagator used in their modeling contains the mathematics of Maxwell’s equations capable of handling the edge boundary conditions. Perhaps I am wrong.

Author Response

Refree:

This is a very interesting paper, which combines theoretical and experimental investigation of potassium doped graphene nanoribbons (GNRs) on a dielectric substrate. First, detailed ab initio calculations of the optical properties of a periodic array of such GNRs are presented for an Al2O3 substrate, which are followed by a comparison of the theory with experiments for electrostatically doped GNRs on a Si/SiO2 substrate that were reported in Ref.[13]. Finally, the authors present their experimental confirmation of the synthesis of GNRs intercalated with K atoms. The paper provides a host of new and valuable information on those systems, so I recommend its publication after the authors consider the following minor comments.

Authors response:

We thank the referee for the positive review, the detailed reading of the manuscript and for very stimulating suggestions regarding the presentation and the parameter choice. We corrected the manuscript correspondingly as much as possible.

Refree:

  1. I think that the presentation of the paper needs more cohesion between its three major parts: (1) ab initio modeling of KC8-NRs on Al2O3, (2) comparison with experiment for electrostatically doped GNR on Si/SiO2 and (3) their own experimental Raman spectra of GNRIC. Namely, while the three parts share much in common, they nevertheless deal with somewhat different systems and/or different processes. I give some examples where a better connection between different parts is desirable.

(a) The doping mechanisms in parts (1) and (2) are different. The electrostatic doping in part (2) may give rise to a rather nonhomogeneous distribution of charge carriers across nanoribbons, whereas the doping by electron donation from the K-atoms should give a more homogeneous distribution. Perhaps the authors can comment on how homogeneous the K-atom density is expected to be across graphene in realistic nanoribbon structures.

Authors response:

In the modeling part (1) we did not pay much attention to the distribution of the doping itself. We imply the homogeneous doping of certain concentration which is a consequence of the homogeneous distribution of the K adatoms. In part (2) we assume that the graphene sample is also homogeneously doped by the holes. In part (3) we know that the intercalated potassium atoms form a 2x2 superlattice, so it is likely that the doping is homogeneous. We are not exactly sure how the doping inhomogeneity would affect the absorption result. We believe that the macroscopic nature of the electromagnetic probe itself would average the effects of inhomogeneous doping.

(b) It is not clear what is achieved by the analysis in part (3) that would be relevant for parts (1) and (2). While part (3) discusses electron-phonon interaction in quite some detail, this interaction is neglected in part (1) and is only briefly mentioned as a possible mechanism that can give rise to a substantial enough broadening of the absorption spectra, which was found to be essential for achieving a good comparison with experiments [13] in part (2).

Authors response:

We thank the referee for this comment and for useful advices.

This is a pioneering study of the electromagnetic modes in heavily doped graphene ribbons that incorporates the retardation effects, the ab initio conductivity tensor calculation and the effects of the substrate polarization through very elegant propagator techniques, which has not been performed tis way so far. The investigation is less focused on the mechanisms of the plasmon damping due to the phonons, because that would lead to additional complications. In the part (2), we used the comparison with the experiment only to test the plausibility of the developed methodology, and this proved to be very successful for a precisely adjusted damping constant. This suggests that the electron phonon interaction (or maybe some other scattering mechanisms) is likely to play a significant role in profiling higher order plasmon resonances n=3,5,.... The referee is right in pointing out that this is poorly emphasized in the paper. In the part (3) we wanted to demonstrate that synthesis of the potassium intercalated graphene ribbons is indeed possible, and to explore how the potassium addatoms modify the strength of the electron-phonon coupling. The latter is very important because the alkali metals can sometimes increase and sometimes decrease the strength of the electrons coupling to the graphene E2g phonon [Nano Lett. 17, 6991 (2017)], which is very important for damping of plasmon resonances, as mentioned. We emphasized this in the new version of the manuscript.

Refree:

  1. I also have a few comments on the choice of modeling parameters. The authors approximate the KC8 as single layer of zero thickness, which is justified in the given range of wavelengths. In other words, they take the distance between graphene and the K-atom layer to be zero. In that respect, I have two questions.

(a) As a result of the above approximation, the two intraband contributions to the conductivity of KC8-SL seem to be fused into a single, strong Drude-lake term with a very large effective number of charge carriers. In view of their modeling of the K(sigma) and C(pi) bands in the inset of Fig.2, is it possible to estimate relative contributions of those two bands in the strong Drude peak?

Authors response:

The referee is right, the spatial dispersivity of the KC8 conductivity in the perpendicular (z) direction is squeezed into one single layer of zero thicknesses so that the two intraband contributions, C(pi) and K(sigma), are fused into a single, strong Drude-like term with a very large effective number of charge carriers. This approximation is fully justified in the optical limit. However, for larger wave vectors Q the perpendicular dispersivity of the dielectric response becomes very import. For example, in that case the C(pi) and K(sigma) behave as two spatially separated 2D plasmas providing Acoustic and Dirac plasmon which we study in Phys. Rev. B 95 201408 (2017). In that publication we also study the C(pi) and Li(sigma) contributions to the effective number of charge carriers. Here we have merged it into one number but we do not know exactly the partial contributions.

(b) The authors keep the distance z0 between the KC8 and the substrate finite (3 or 4 Angstroms). It seems to me that taking z0 = 0 would be consistent with the zero-thickness approximation for the KC8-SL in the given range of wavelengths. The distance between graphene layer and the K-atom layer is on the order of 3 Angstroms, so if that is neglected, why not taking z0 = 0?

Authors response:

We thank the referee for pointing out this relevant point. The reason why we insist on the correct z0 is following. In the optical limit Q\rightarrow 0, \beta_0z_0\approx 0 and the exponential factor exp(i\beta_0z_0) in the scattered propagator Eq.15 is, as the referee noted, independent on z_0, so, we could take z_0=0 instead of 3 or 4 Angstroms. However, when we introduce the ribbons, the Bragg scattering occurs (Q\rightarrow Q+G) and the scattered propagator gains the reciprocal vector G (Eq.10). Moreover, we noticed that we have to include large number of G vectors (more than a hundred) in order to converge results. As a result, the maximum wave vectors Q+G are not so small any more (they can be up to ~0.1a.u.) which causes the propagator Eq.15 to become, weakly but still noticeably, sensitive to the choice of z0. We hope that this explanation is satisfactory.

Referee:

  1. My final comment refers to the very interesting remark on the observed splitting of the n=2 dark plasmon mode in the ab initio spectra in Fig.7. The authors state that “This may be the evidence of the surface or edge plasmons localized at the KC8-NR boundaries [15].” I am not entirely convinced that the electric field propagator used in their modeling contains the mathematics of Maxwell’s equations capable of handling the edge boundary conditions. Perhaps I am wrong.

Authors response:

Again, we thank the referee for this very relevant comment. We are not sure that an edge plasmon actually appears in the Fig.7. However, we are sure that the propagator technique we presented is analogous to solving the Maxwell's equations for any planar boundary conditions. So, regardless of the shape of the edge, sharp or atomically smooth, the screened conductivity provides the exact current or charge fluctuations at the edges and finally the exact electromagnetic field. The problem may be the ill-defined boundary conditions such as lines at the 2D edges, which is the situation studied here. Then the calculation requires large number of the Fourier components (gg’) in order to properly expand the discontinuity at the edges. However, the induced density at the ribbon edges in Fig.5a makes it clear that our method copes well with this problem.

Reviewer 6 Report

The manuscript comprises interesting results that might be of high interest for graphene community. It is not clear however, how novel the theoretical part is. Could you please clarify the updated provided by your data vs space-time dispersion of graphene conductivity [The European Physical Journal B volume 56, pages281–284 (2007)], as well as inter- and intra-band graphene conductivity calculated by Sergey Mikhailov.

It also makes sense to compare the absorption provided by graphene ribbons with the THz absorption of graphene/polymer multilayers as they are and biased by IR pulses (see Appl.Phys. Lett. 108, 123101 (2016) and ACS Photonics, 2019, 6 (3), pp 720–727).

Author Response

Refree:

The manuscript comprises interesting results that might be of high interest for graphene community. It is not clear however, how novel the theoretical part is. Could you please clarify the updated provided by your data vs space-time dispersion of graphene conductivity [The European Physical Journal B volume 56, pages281–284 (2007)], as well as inter- and intra-band graphene conductivity calculated by Sergey Mikhailov.

It also makes sense to compare the absorption provided by graphene ribbons with the THz absorption of graphene/polymer multilayers as they are and biased by IR pulses (see Appl.Phys. Lett. 108, 123101 (2016) and ACS Photonics, 2019, 6 (3), pp 720–727).

Authors response:

We thank the referee for the positive review, for the detailed reading of the manuscript and for pointing out this interesting and relevant publications. We corrected the manuscript in accordance and introduced all suggested references.

Regarding the novelty of the theoretical part we can say the following.

This is a pioneering formulation of the electromagnetic modes in heavily doped graphene ribbons that incorporates the retardation effects, the ab initio conductivity tensor calculation and the effects of the substrate polarization through very elegant propagator techniques, which has not been performed this way so far. The investigation is less focused on the mechanisms of the plasmon damping due to the phonons, because that would lead to additional complications. We provided the comparison with the experiment in order to test the plausibility of the developed methodology, and this proved to be very successful for a precisely adjusted damping constant. This suggests that the electron phonon interaction (or maybe some other scattering mechanisms) is likely to play a significant role in profiling the higher order plasmon resonances n=3,5,.... The conductivity tensor in the KC8 single layer is calculated from first principles including all different intraband and interband excitation channels. The lack of the presented theoretical description of conductivity is that the spatial dispersion of the KC8 conductivity in the perpendicular (z) direction is reduced to a single layer of zero thickness so that the two intraband contributions, C(pi) and K(sigma), are fused into a single, strong Drude-like term with a very large effective number of charge carriers. This approximation is fully justified in the optical limit. However, for larger wave vectors Q the perpendicular dispersivity of the dielectric response becomes very import. For example, in that case the C(pi) and K(sigma) behave as two spatially separated 2D plasmas providing Acoustic and Dirac plasmon which we study in Phys. Rev. B 95 201408 (2017). In the suggested Eur. Phys. J. B 56, 281–284 (2007) the graphene conductivity is treated in Dirac cone approximation (two conical bands). This approximated graphene conductivity would probably also provide the interesting phenomenon studied in our paper, however, it should be taken into account that we are studying KC8, which also has extra K(sigma) band crossing the Fermi level. The proposed publications Appl.Phys. Lett. and ACS Photonics present the enhanced THz absorption of the graphene multilayers on various dielectric substrates which is a very interesting and useful topic, so we cited them in the revised manuscript.
